## Systematic review protocol of the effectiveness of HIV prevention interventions for reducing risky sexual behaviour among youth globally

Fungai Mbengo,[1] Maggie Zgambo,[1] Ebenezer Afrifa-Yamoah [ID] ,[2] Fatch Welcome Kalembo,[3] Takanori Honda [ID] ,[4] Yoko Shimpuku,[5] Sanmei Chen [ID] [5]

For numbered affiliations see end of article.

**Correspondence to**
Dr Sanmei Chen;
chens@hiroshima-u.ac.jp

## ABSTRACT

**Introduction** Human immunodeficiency virus (HIV) prevention interventions focused at reducing risky sexual behaviours are an important strategy for preventing HIV infection among youth (15–24 years) who continue to be vulnerable to the disease. This systematic review aims to synthesise current global evidence on the effectiveness of HIV prevention interventions for reducing risky sexual behaviour among youth in the last decade.

**Methods and analysis** MEDLINE/PubMed, EMBASE, PsychINFO, ProQuest Central, CINAHL and Web of Science databases, ClinicalTrials.gov and the WHO International Clinical Trials Registry Platform and reference lists of included studies and systematic reviews on effectiveness of HIV prevention interventions for reducing risky sexual behaviour among youth will be searched for articles published from August 2011 to August 2021. Eligible studies will be longitudinal studies including randomised controlled trials and quasi-experimental studies that examined the effectiveness of HIV prevention interventions among youth populations (15–24 years) with risky sexual behaviour as a primary or secondary outcome. Study selection and quality assessment will be undertaken independently by three reviewers and disagreements will be resolved through consensus. Data analysis will be undertaken using RevMan software V.5.3.3. A random effects meta-analysis will be conducted to report heterogeneous data where statistical pooling is achievable. We will use $I^2$ statistics to test for heterogeneity. Where appropriate, a funnel plot will be generated to assess publication bias. Where statistical pooling is unachievable, the findings will be reported in a narrative form, together with tables and figures to assist in data presentation if required. Reporting of the systematic review will be informed by Preferred Reporting Items for Systematic Reviews and Meta-Analyses (PRISMA) guidelines.

**Ethics and dissemination** Ethical approval is not required. Findings of the systematic review will be published in a peer-reviewed journal. The findings will be of interest to researchers, healthcare practitioners and policymakers.

**PROSPERO registration number** CRD42021271774.

## STRENGTHS AND LIMITATIONS OF THIS STUDY

⇒ This systematic review protocol follows the Preferred Reporting Items for Systematic Review and Meta-analysis Protocols guidelines with transparency to the methods and processes that will be used.
⇒ This systematic review will address a gap in the current global evidence by comprehensively including research findings worldwide on the effectiveness of HIV prevention interventions for reducing risky sexual behaviour among youth in the last decade.
⇒ This systematic review will be one of the few to focus on behavioural-level, structural-level and combined interventions for HIV prevention targeting youth.
⇒ This systematic review will consider only publications written in English, which will result in language bias.
⇒ The measures of risky sexual behaviour may vary between studies, which may create bias and render pooled of estimates meaningless.

## INTRODUCTION

According to the Joint United Nations Programme on human immunodeficiency virus (HIV)/acquired immunodeficiency syndrom (AIDS) (UNAIDS), 38 million people were living with HIV in 2019 and about 35 million had died since HIV was first recognised globally.[1] Over the last decade, there has been a remarkable reduction in HIV infections, HIV related mortality and morbidity in the general population globally following the introduction of the universal antiretroviral therapy.[2] However, recent reports indicate that HIV remains a leading cause of death among the youth population across Sub-Saharan Africa, where 80% of young people living with HIV reside.[1 3] Furthermore, in 2019, one third of the global HIV infections occurred among youth aged between 15 and 24 years and the infection

rate in this population is anticipated to increase yearly by 13% leading to an estimation of 3.5 million new infections by 2030.[4 5]

Youth are vulnerable to HIV because of a myriad of factors that influence risky sexual behaviours including low self-esteem, poverty, peer pressure,[6] alcohol or drug abuse,[7] cultural practices, gender-disparities,[8 9] limited HIV-related knowledge and gender-based violence.[10] According to the socio-ecological model, these factors affect youth at individual, interpersonal, community, institutional and structural levels.[11 12] This model has been used to guide the development of HIV prevention interventions aimed at reducing risky sexual behaviours among youth.[12–15] In this review, risky sexual behaviours are activities engaged by an individual that result in negative outcome and these activities are engaging in transactional sex, unprotected sex, multiple sex partners, intergenerational sex, having partners who are at risk of HIV infection such as using injectable drugs, early sex debut and early marriages.[16–18]

HIV prevention interventions for reducing risky sexual behaviours can be classified as either structural, behavioural or combined.[13–15] Structural interventions aim at reducing risky sexual behaviour by addressing structural-level factors such as poverty, unemployment, limited access to education and social norms.[13–15] The behavioural interventions facilitate risky sexual behaviour reduction by targeting individual-level factors such as knowledge, attitudes, self-esteem and self-efficacy.[13–15] Lastly, the combined interventions reduce risky sexual behaviour by addressing one or more individual-level factors, as well as one or more structural-level factors.[13–15]

In the last decade, researchers have conducted systematic reviews on structural, behavioural or combined HIV prevention interventions targeting youth.[13 19–24] However, these reviews have synthesised research findings from studies that were conducted in a specific region, for instance, Sub-Saharan Africa,[13 19 20] developing countries,[21 22] middle-income countries,[23] or a specific country.[24] One review has particularly focused on the female gender.[13] Most of these reviews have focused on behavioural interventions only.[19 21–23] Synthesising global evidence on the subject has potential to identify more effective interventions for reducing risky sexual behaviours among youth.

Following this background, we propose to undertake a systematic review and meta-analysis to synthesise current global evidence on the effectiveness of HIV prevention interventions for reducing risky sexual behaviours among youth in the last decade. Specifically, this systematic review and meta-analysis seeks to answer the following questions:1) What HIV prevention interventions are effective for reducing risky sexual behaviour among youth? 2) What is the level of effectiveness of interventions designed to reduce risky sexual behaviour among youth? 3) What factors affect the effectiveness of the identified interventions? To prevent duplication of reviews, a preliminary search of similar protocols or reviews was conducted in July 2021 in CINAHL, Cochrane Library, MEDLINE, Google Scholar, The International Prospective Register of Systematic Reviews (PROSPERO) and Joanna Briggs Institute (JBI) databases. No review protocol or systematic review on this topic published in the last decade was identified.

## METHODS AND ANALYSIS

This will be a systematic review and meta-analysis designed to analyse recent global evidence on the effectiveness of HIV prevention interventions for reducing risky sexual behaviour among youth. This review will commence in April 2022. This protocol has been developed in accordance with the Preferred Reporting Items for Systematic Review and Meta-analysis Protocols (PRISMA-P) guidelines.[25] Reporting of the synthesised findings will be informed by PRISMA guidelines.[26] This protocol has been registered in the PROSPERO. Important amendments to this protocol will be published along with the results of the systematic review.

### Eligibility criteria

To be eligible for this review, studies will need to report on youth aged from 15 to 24 years and be conducted in either clinical or non-clinical setting. Inclusion of studies with participants aged within and outside the identified age bracket will depend on the mean age that falls within the age range of 15–24 years. For recency, all studies conducted from August 2011 to August 2021 will be considered. The review will include studies conducted in any geographical location. This review will include the following three types of interventions: (1) behavioural interventions that facilitate reduction of risky sexual behaviour by addressing individual-level factors, (2) structural interventions that facilitate reduction of risky sexual behaviour by targeting structural-level factors, (3) combined interventions that facilitate reduction of risky sexual behaviour by targeting at least one individual-level factor and at least one structural-level factor. Studies that do not measure risky sexual behaviour as a primary or secondary outcome will be excluded. In the source trials, the comparators could be control groups who received no intervention or alternative usual interventions or wait-list controls. The review will consider only longitudinal studies such as quasi-experimental studies, randomised controlled trials and cluster-randomised trials and non-randomised controlled trials. Only publications written in English will be considered.

### Information sources

We will electronically search the following databases to retrieve relevant articles: MEDLINE/PubMed, EMBASE, PsychINFO, ProQuest Central, CINAHL and Web of Science databases. We will also search ClinicalTrials.gov and the World Health Organisation (WHO) International Clinical Trials Registry Platform to identify ongoing or unpublished eligible trials. To maximise the search

**Table 1** Search grid with identified PICO concepts

| PICO concepts | |
|---|---|
| Participants | Youth aged from 15 to 24 years, or the mean age of participants falls in the age range of 15–24 years |
| Intervention | HIV prevention intervention programme aimed at reducing risky sexual behaviour, including structural, behavioural and combined |
| Comparators | Control groups who received no intervention or alternative usual interventions, or waitlist controls. |
| Outcomes | Risky sexual behaviours: (1) having first sexual activity at or before the age of 15 (early sexual debut), (2) engaging in sexual activity without a condom, (3) inconsistent condom use, (4) having multiple sex partners, (5) intergenerational sex, (6) transactional sex, (7) forced sex, (8) early marriages, (9) having partners who are at risk of HIV infection sich as using injectable drugs and (10) having been pregnant or fathered a child at a younger age (18 years or younger). |

PICO, Population, Intervention, Comparator and Outcome.

for relevant articles, we will review reference lists of included studies and systematic reviews on effectiveness of HIV prevention interventions for reducing risky sexual behaviour among youth.

## Search strategy

The Population, Intervention, Comparator and Outcome (PICO) Model as recommended by JBI will be used to develop a search grid for this review. In this review, the participants are youth aged from 15 to 24 years of age; interventions are HIV prevention interventions; comparators are control groups, and outcomes are risky sexual behaviours. The search grid with identified PICO concepts is presented in table 1.

Using the identified PICO concepts, a three-step search strategy will be utilised to identify relevant studies. First, keywords for PICO concepts will be brainstormed by reviewers before undertaking an initial limited search of PICO concepts in the MEDLINE/PubMed database to identify controlled vocabulary Medical Subject Headings (MeSH). A second search of EMBASE, MEDLINE/PubMed, PsychINFO, ProQuest Central, CINAHL and Web of Science databases, ClinicalTrials.gov and the WHO International Clinical Trials Registry Platform will be conducted using the identified keywords and MeSH terms. Third, we will use these information sources to find other relevant keywords and terms. Boolean operators such as 'OR' and 'AND' will be applied when combining similar search terms and different search terms, respectively. For example, (youth OR young people OR teen OR young adults OR adolescents) AND (HIV prevention intervention OR HIV prevention strateg* OR HIV prevention program* OR HIV education prevention program* OR Sexual* education program*). A detailed search strategy is presented in online supplementary file table

S1, using MEDLINE/PubMed as an example. The search strategy will be adapted to other information sources.

## Study selection

Following the search, all identified citations will be collated and uploaded into EndNote V.X9 Reference Management System. Duplicates will be removed before importing the references into the Covidence online systematic review tool as recommended by the Cochrane Collaboration.[27] This tool is designed to assist reviewers to screen abstracts and full texts of identified articles, risk of bias assessment and data extraction.[27] In this review, however, this tool will only be used to select eligible studies. Abstracts of the relevant full texts will be assessed for eligibility by FM, MZ and SC, independently. Full-text articles for the selected titles will be further reviewed independently by these reviewers. Following this selection, methodological quality of each included study will be independently assessed by three reviewers using the JBI Critical Appraisal Checklist for Randomized Controlled Trials and JBI Critical Appraisal Checklist for Quasi-Experimental Studies.[28] These two appraisal tools include 12 and 9 criteria (rating: yes, no, unclear), respectively, with a narrative form for decision-making. Two reviewers will rate each study as a JBI score ranging from 1 to the total score (12 or 9), with higher score indicating higher quality. Studies that meet less than 50% of all criteria will be excluded. Each reviewer will leave an audit trail with reasons for each decision undertaken, which will later be compared across all the reviewers. Cohen's kappa coefficient will be used to measure agreement among the reviewers. Any disagreements that arise among the reviewers at each stage of the study selection process will be resolved through discussions to reach a consensus. The results of the search will be illustrated following the PRISMA flow diagram.[26]

## Assessment of risk of bias and quality of evidence

As mentioned above, three reviewers will independently rate the quality of each included study using the JBI Critical Appraisal Checklist for Randomized Controlled Trials and JBI Critical Appraisal Checklist for Quasi-Experimental Studies.[28] In addition, we will examine the quality of evidence for each outcome by using the Grading of Recommendations, Assessment, Development and Evaluations (GRADE) approach,[29] because the quality of evidence often varies between outcomes.[30] We will not exclude any study on the basis of the GRADE score. Authors of papers will be contacted to request missing information that is necessary for appraisal.

## Data extraction

Data will be extracted by FM using the standardised JBI data extraction tool.[28] MZ and SC will verify the extracted data. Details extracted will include: (1) characteristics of the study: author, year of publication, the title of the study, study objective, study location, setting; (2) methodological characteristics: study design, research questions

and/or hypotheses, study population, sample characteristics, groups and controls, type of intervention, length of intervention, delivery mode, theoretical framework and length of follow-up, measurements, data analyses; (3) main findings and conclusions. Authors will be contacted to request for full articles if only abstracts are accessible and for information if the main outcome data and methods are missing or unclear.

## Outcomes and prioritisation

The primary outcome of this review is risky sexual behaviour, which will be defined as engaging in one or more of the following sexual behaviours as derived from the literature: (1) having first sexual activity at or before the age of 15 (early sexual debut), (2) engaging in sexual activity without a condom, (3) inconsistent condom use, (4) having multiple sex partners, (5) intergenerational sex, (6) transactional sex, (7) forced sex, (8) early marriages, (9) having partners who are at risk of HIV infection such as using injectable drugs and (10) having been pregnant or fathered a child at a younger age (18 years or younger).[17 31 32] The secondary outcome measures will be sexually transmitted diseases rates, pregnancy rates, birth rates and changes in mediating factors that affect risky sexual behaviours such as knowledge, attitudes and beliefs. Interventions will be deemed effective if the frequency of behaviour is reduced or stopped and if there is a positive change in the prevalence of the secondary outcomes and mediating factors. Factors that affect the effectiveness of the intervention such as study design, sample characteristics, intervention content, duration of intervention, the dosage of intervention, length of follow-up and theoretical framework will also be examined.

## Data synthesis

Statistical analyses will be conducted using RevMan software (V.5.3.3; The Cochrane Collaboration). We will perform analyses for all outcomes on an intention to treat basis. Effect sizes expressed as OR (for dichotomous data) and weighted mean differences or standardised mean differences (for continuous data) and their 95% CIs will be calculated for analysis. A $p$ value of less than 0.05 will be considered as statistically significant. Heterogeneity will be assessed statistically using the $\chi^2$ test and the $I^2$ index.[33] We will use fixed effects models to pool outcomes if significant heterogeneity is not present ($I^2 < 50\%$). We will use random effects models when significant heterogeneity is present ($I^2 \geq 50\%$). If sufficient data are available, subgroup analyses will be conducted to explore the heterogeneity between the studies. A funnel plot will be generated to assess publication bias if there are 10 or more studies included in a meta-analysis. Statistical tests for funnel plot asymmetry (Egger test, Begg test, Harbord test) will be performed where appropriate. We will analyse the data separately for

each of the three types of interventions (ie, structural, behavioural or combined interventions). Several subgroup analyses will be conducted based on study setting, year of publication, length of follow-up and so on. We will also conduct sensitivity analyses to test the robustness of our findings, such as by excluding quasi-randomised trials. Where statistical pooling is unachievable,[34] the findings will be reported in a narrative form, together with tables and figures to assist in data presentation if required.

## Patient and public involvement

No patient involved.

## ETHICS AND DISSEMINATION

Ethical approval is not required for a systematic review. Findings of the systematic review will be disseminated through publication in a peer-reviewed journal. The findings will be of interest to researchers, healthcare practitioners and policymakers.

**Author affiliations**
[1]School of Nursing and Midwifery, Edith Cowan University, Joondalup, Perth, Australia
[2]School of Science, Edith Cowan University, Perth, Western Australia, Australia
[3]School of Nursing, Midwifery and Paramedicine, Curtin University, Perth, Western Australia, Australia
[4]Department of Epidemiology and Public Health, Graduate School of Medical Sciences, Kyushu University, Fukuoka, Japan
[5]Global Health Nursing, Department of Health Sciences, Graduate School of Biomedical and Health Sciences, Hiroshima University, Hiroshima, Japan

**Contributors** FM, MZ and SC conceptualised and designed the protocol, and drafted the manuscript. EA-Y, FWK, TH and YS contributed to the plan of search strategy, data extraction and data analysis. All authors contributed to the manuscript revision.

**Funding** SM was supported by Grants-in-Aid for Early-Career Scientists (JP19K19474) from the Ministry of Education, Culture, Sports, Science and Technology of Japan.

**Disclaimer** The funding source had no role in the study conception, the preparation of the manuscript, or the decision to submit the manuscript for publication.

**Competing interests** None declared.

**Patient and public involvement** Patients and/or the public were not involved in the design, or conduct, or reporting, or dissemination plans of this research.

**Patient consent for publication** Not applicable.

**Provenance and peer review** Not commissioned; externally peer reviewed.

and indication of whether changes were made. See: https://creativecommons.org/licenses/by/4.0/.

**ORCID iDs**

Ebenezer Afrifa-Yamoah http://orcid.org/0000-0003-1741-9249
Takanori Honda http://orcid.org/0000-0002-1011-9879
Sanmei Chen http://orcid.org/0000-0003-0811-1701

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
