## [Reviewer comments · BMJ Open]

ARTICLE DETAILS

TITLE (PROVISIONAL)	A systematic review protocol of the effectiveness of HIV prevention interventions for reducing risky sexual behaviour among youth globally
AUTHORS	Mbengo, Fungai; Zgambo, Maggie; Afrifa-Yamoah, Ebenezer; Kalembo, Fatch; Honda, Takanori; Shinpuku, Yoko; Chen, Sanmei

VERSION 1 – REVIEW

REVIEWER	Minttu Rönn Harvard T. H. Chan School of Public Health, Department of Global Health and Population
REVIEW RETURNED	02-Dec-2021

GENERAL COMMENTS	The protocol titled: “A systematic review protocol of the effectiveness of HIV prevention interventions for reducing risky sexual behaviour among youth globally” is well written, and it clearly outlines the plan for the systematic review to synthesize global evidence on effectiveness of interventions aimed at reducing risky sexual behavior among youth. 1. My main comment relates to the measurement of baseline levels of sexual risk, and how these may vary within the study population and between the studies. What is defined as risky sexual behavior, and the prevalence of risky sexual behaviors is likely to vary greatly, and the baseline measures (as well as follow-up measures) may be biased. The variation, and biases, may render pooled estimates meaningless, and I am interested to hear how the authors have conceptualized these challenges and how they plan to address them in their systematic review and in data analysis. Otherwise, I have very minor comments. 2. In the section of “Assessment of risk of bias and quality of evidence” the authors mention 50% quality threshold as an inclusion criteria. This seems a little arbitrary at first glance and there is no mention of how the quality scale is constructed. Having more transparent criteria presented would be useful to understand what types of studies would not qualify for inclusion based on the quality threshold. 3. At the end of data analysis section: “We will also conduct sensitivity analyses to test the robustness of our findings, such as by excluding quasi-randomised trials, excluding trials with high or unknown risk of bias.” This seems to contradict the inclusion if only high-quality studies are included, which should have lower risk of bias (comment #2). Can you clarify what is meant with this? 4. “Only studies conducted in English will be considered”. I assume you mean publications written in English, instead of only considering studies where the language of the study was English? Consider
---

	rephrasing, and if the intention is the latter, what is the rationale for this? 5. I assume you will only include longitudinal studies (by inclusion of trials) but it would be good to mention it as an explicit inclusion criteria.
--	--

VERSION 1 – AUTHOR RESPONSE

Reviewer 1

Comment #1 My main comment relates to the measurement of baseline levels of sexual risk, and how these may vary within the study population and between the studies. What is defined as risky sexual behavior, and the prevalence of risky sexual behaviors is likely to vary greatly, and the baseline measures (as well as follow-up measures) may be biased. The variation, and biases, may render pooled of estimates meaningless, and I am interested to hear how the authors have conceptualized these challenges and how they plan to address them in their systematic review and in data analysis.

Response 1: *Thank you for your comment. Indeed the definition of sexual risk behaviour may vary among the included studies. In our review, we will follow the definition of sexual behaviours provided on page 13 of the manuscript. An adolescent will be considered to have engaged in risky sexual behaviour if they have engaged in one or more of the 10 risky sexual behaviours provided in page 13. These behaviours have been outsourced from the literature. While we understand that some studies may only focus on a specific behaviour, understanding of prevalence will base on a specific behaviour, for instance, prevalence of condomless sex. The prevalence will be presented in ranges or means. We also anticipate discovering some risky sexual behaviours that are not listed in our protocol and we plan to include these in the analysis. For this reason, we will not restrict our definition of sexual risk behaviours to the 10 behaviours mentioned in the protocol. Depending on the heterogeneity of risk of bias in the included studies, robustness of our findings will be tested by undertaking sensitivity analyses (analysing studies with higher risk and lower risk bias separately). We rephrased this section as follows:*

“The primary outcome of this review is risky sexual behaviour, which will be defined as engaging in one or more of the following sexual behaviours as derived from literature: 1) having first sexual activity at or before the age of 15 (early sexual debut), 2) engaging in sexual activity without a condom, 3) inconsistent condom use, 4) having multiple sex partners, 5) intergenerational sex, 6) transactional sex 7) forced sex, 8) early marriages, 9) having partners who are at risk of HIV infection such as using injectable drugs and 10) having been pregnant or fathered a child at a younger age (18 years or younger).^{17,31,32} The secondary outcome measures will be sexually transmitted diseases (STDs) rates, pregnancy rates, birth rates and changes in mediating factors that affect risky sexual behaviours such as knowledge, attitudes and beliefs. Interventions will be deemed effective if the frequency of behaviour is reduced or stopped and if there is a positive change in the prevalence of the secondary outcomes and mediating factors.” (Page 13)

We accordingly revised the outcome in PICO concepts as shown in Table 1 (Page 15).

We rephased relevant section in the Introduction:

“In this review, risky sexual behaviours are activities engaged by an individuals that result in negative outcome and these activities are engaging in transactional sex, unprotected sex, multiple sex partners, intergenerational sex, having partners who are at risk of HIV infection such as using injectable drugs, early sex debut and early marriages.^{16,17,18”} (Page 6–7)

Comment #2 In the section of “Assessment of risk of bias and quality of evidence” the authors mention 50% quality threshold as an inclusion criteria. This seems a little arbitrary at first glance and there is no mention of how the quality scale is constructed. Having more transparent criteria presented would be useful to understand what types of studies would not qualify for inclusion based on the quality threshold.

Response 2: *Thank you for bringing us this important issue. As per your suggestion, to be more transparent, we clarified selection criteria in the section of “Study Selection” as follows:*

“Following this selection, methodological quality of each included study will be independently assessed by three reviewers using the JBI Critical Appraisal Checklist for Randomized Controlled Trials and JBI Critical Appraisal Checklist for Quasi-Experimental Studies.²⁸ These two appraisal tools include 12 and 9 criteria (rating: yes, no, unclear), respectively, with a narrative form for decision making. Two reviewers will rate each study as a JBI score ranging from 1 to the total score (12 or 9), with higher score indicating higher quality. Studies that meet less than 50% of all criteria will be excluded. Each reviewer will leave an audit trail with reasons for each decision undertaken, which will later be compared across all the reviewers. Cohen’s kappa coefficient will be used to measure agreement among the reviewers.” (Page 11–12)

Since we will utilise the GRADE tool only to assess quality in the included studies for each outcome measure. We further explained it as follows:

“We will not exclude any study on the basis of the GRADE score.” (Page 12)

Comment #3 At the end of data analysis section: “We will also conduct sensitivity analyses to test the robustness of our findings, such as by excluding quasi-randomised trials, excluding trials with high or unknown risk of bias.” This seems to contradict the inclusion if only high-quality studies are included, which should have lower risk of bias (comment #2). Can you clarify what is meant with this?

Response 3: *Thank you for your comment. As you pointed out, there is no need to do a sensitivity analysis of excluding trials with high risk of bias. Accordingly, we removed it from this section. Now it reads like:*

“We will also conduct sensitivity analyses to test the robustness of our findings, such as by excluding quasi-randomised trials.” (Page 14)

Comment #4 “Only studies conducted in English will be considered”. I assume you mean publications written in English, instead of only considering studies where the language of the study was English? Consider rephrasing, and if the intention is the latter, what is the rationale for this?

Response 4: *Thank you for your suggestion. We mean publications written in English. We have rephrased this as follows:*

“Only publications written in English will be considered”. (Page 9)

Comment #5 I assume you will only include longitudinal studies (by inclusion of trials) but it would be good to mention it as an explicit inclusion criteria.

Response 5: *Thank you for your suggestion. Yes, we will include only longitudinal studies and we have mentioned this as an explicit inclusion criteria. We have revised this to:*

“The review will consider only longitudinal studies such as quasi-experimental studies, randomized controlled trials and cluster-randomized trials and non-randomized controlled trials.”
(Page 9)